# Evaluation of BAFF, APRIL and CD40L in Ocrelizumab-Treated pwMS and Infectious Risk

**DOI:** 10.3390/biology12040587

**Published:** 2023-04-12

**Authors:** Maria Antonella Zingaropoli, Patrizia Pasculli, Matteo Tartaglia, Federica Dominelli, Federica Ciccone, Ambra Taglietti, Valentina Perri, Leonardo Malimpensa, Gina Ferrazzano, Marco Iannetta, Cosmo Del Borgo, Miriam Lichtner, Claudio Maria Mastroianni, Antonella Conte, Maria Rosa Ciardi

**Affiliations:** 1Department of Public Health and Infectious Diseases, Sapienza University of Rome, 00185 Rome, Italy; 2Department of Human Neurosciences, Sapienza University of Rome, 00185 Rome, Italy; 3Infectious Disease Unit, Department of System Medicine, Tor Vergata University and Hospital, 00133 Rome, Italy; 4Infectious Diseases Unit, Santa Maria Goretti Hospital, Sapienza University of Rome, 04110 Latina, Italy; 5Department of Neurosciences Mental Health and Sensory Organs, Sapienza University of Rome, 00185 Rome, Italy; 6IRCCS Neuromed, 86077 Pozzilli, Italy

**Keywords:** pwMS, ocrelizumab, DMTs, anti-CD20, BAFF, APRIL, CD40L, infectious risk

## Abstract

**Simple Summary:**

Since B cells have been linked to multiple sclerosis (MS) and its progression as well as T cells, the second-generation anti-CD20 recombinant humanized monoclonal antibody ocrelizumab has been approved for MS treatment. Although ocrelizumab efficiently depletes B cells in peripheral blood, some B cells and CD20 negative plasma cells persist in lymphatic organs, and their survival is regulated by the B-cell-activating factor (BAFF)/a proliferation-inducing ligand (APRIL) system. Moreover, ocrelizumab may result in higher infectious risk. Herein, we investigated plasma BAFF, APRIL and CD40L levels and their relationship with infectious risk in ocrelizumab-treated people with (pw) MS at baseline, at 6 months and at 12 months after starting the treatment, comparing the above-mentioned findings with a control group. At baseline, plasma levels of all three cytokines were higher compared to the control group. In pwMS, the longitudinal assessment showed a significant increase in plasma BAFF levels and a significant reduction in plasma APRIL and CD40L. Moreover, when stratifying pwMS according to the onset of an infectious event during the 12-month follow-up period, significantly higher plasma BAFF levels were found at all time-points in the group with an infectious event than in the group without an infectious event. Hence, BAFF may have a role as a marker of immune dysfunction and infectious risk.

**Abstract:**

Background: The anti-CD20 monoclonal antibody ocrelizumab has been widely employed in the treatment of people with multiple sclerosis (pwMS). However, its B-cell-depleting effect may induce a higher risk of infectious events and alterations in the secretion of B-cell-activating factors, such as BAFF, APRIL and CD40L. Methods: The aim of this study was to investigate plasma BAFF, APRIL and CD40L levels and their relationship with infectious risk in ocrelizumab-treated pwMS at baseline (T0), at 6 months (T6) and at 12 months (T12) after starting the treatment. As a control group, healthy donors (HD) were enrolled too. Results: A total of 38 pwMS and 26 HD were enrolled. At baseline, pwMS showed higher plasma BAFF (*p* < 0.0001), APRIL (*p* = 0.0223) and CD40L (*p* < 0.0001) levels compared to HD. Compared to T0, plasma BAFF levels were significantly increased at both T6 and T12 (*p* < 0.0001 and *p* < 0.0001, respectively). Whereas plasma APRIL and CD40L levels were decreased at T12 (*p* = 0.0003 and *p* < 0.0001, respectively). When stratifying pwMS according to the development of an infectious event during the 12-month follow-up period in two groups—with (14) and without an infectious event (24)—higher plasma BAFF levels were observed at all time-points; significantly, in the group with an infectious event compared to the group without an infectious event (T0: *p* < 0.0001, T6: *p* = 0.0056 and T12: *p* = 0.0400). **Conclusions:** BAFF may have a role as a marker of immune dysfunction and of infectious risk.

## 1. Introduction 

Multiple sclerosis (MS) is one of the most frequent neurological disorders affecting young adults in the world, characterized by widespread presence of inflammatory foci in the central nervous system (CNS) accompanied by focal demyelination and glial scarring in particular in the initial relapsing–remitting phase in which new neurological symptoms occur with or without disability accumulation [1]. Accumulation of focal and diffuse damage in both white and gray matter with progressive disability characterizes the progressive phase of the disease [2,3]. In progressive MS, chronic inflammation persists in both parenchymal and meningeal spaces, being at least partly compartmentalized across a relatively intact blood–brain barrier [4,5]. This phase is associated with neurodegeneration that strengthens over time, thus leading to irreversible tissue damage [6].

The growing understanding of the immunopathogenesis of multiple sclerosis (MS) has led to the development of innovative treatments, and disease-modifying therapies (DMTs) have become the gold standard. [7]. According to numerous experimental animal model studies, the chronic inflammation associated with MS was primarily considered a T-cell-mediated process. However, B cells have been linked to MS and its progression as well as T cells [8], and the second-generation anti-CD20 recombinant humanized monoclonal antibody (mAb) ocrelizumab has been approved for the treatment of both relapsing–remitting MS (RRMS) and primary progressing MS (PPMS) [7,9,10,11]. It selectively induces B cell depletion, resulting in an immunosuppressive effect [12,13,14,15]. Although ocrelizumab efficiently depletes B cells in blood, some B cells and CD20 negative plasma cells persist in lymphatic organs and the inflamed central nervous system (CNS). Their survival is regulated by the B-cell-activating factor (BAFF)/a proliferation-inducing ligand (APRIL) system. As B cell activation markers, BAFF, APRIL and CD40 ligand (CD40L) have been implicated in the persistence of self-reactive immune responses and may have a proinflammatory role [16,17,18,19,20,21].

The role of B cells in MS is not fully understood. Clinical trials had demonstrated a dual function of B cells in MS. Indeed, B cells have both inflammatory and anti-inflammatory functions in MS. It is generally believed that B cells contribute to MS by producing autoantibodies, expressing inflammatory cytokines and presenting antigens to T helper cells [22]. The anti-CD20 clinical trials in people with MS (pwMS) provide evidence regarding the inflammatory function of B cells [23]. However, drugs targeting the BAFF/APRIL system that have been adopted for the treatment of other autoimmune disorders have, to date, provided unsatisfactory results in MS. Indeed, clinical trials with atacicept, a recombinant fusion protein that suppresses B cell function and proliferation, blocking BAFF and APRIL, increased disease activity in MS, demonstrating that some B cell subsets may have anti-inflammatory functions [24]. Although depleting B cells with anti-CD20 antibodies is effective in treating MS, atacicept treatment, blocking BAFF and APRIL, paradoxically increases disease activity in pwMS. The reason behind the failure of atacicept is not well understood.

Finally, despite its remarkable efficacy, ocrelizumab is associated with an increased risk of infection and reactivation [12,25,26,27,28,29,30,31,32,33]. Considering the intense B cell depletion induced by ocrelizumab, the aim of this study was to investigate the alterations in B cell activation markers prior to and during ocrelizumab treatment and their relationship with the occurrence of infectious events.

## 2. Materials and Methods

### 2.1. Study Participants

This study was evaluated and approved by the Ethics Committee of Policlinico Umberto I, Sapienza University of Rome (protocol numbers 130/13 and 353/20). Study participation was conditional upon written informed consent.

As previously described [31], at the Policlinico Umberto I Neuroinfectious Unit, people with MS (pwMS) were evaluated and enrolled. Specifically, in the context of a collaboration between the MS Centre and the Neuroinfectious Unit, all pwMS are evaluated every six months to recognize new or reactivation of latent infections.

MS diagnosis was based on the 2017 McDonald criteria, and pwMS neurological disability was assessed according to Expanded Disability Status Scale (EDSS) score.

### 2.2. Sample Collection

As reported in Figure 1, ocrelizumab was administered on a 6-month schedule. To evaluate the plasma levels of B cell activation markers in ocrelizumab-treated pwMS, peripheral blood samples were routinely collected in ethylenediaminetetraacetic acid (EDTA) tubes from pwMS at three time-points: before the administration of the first (T0), the second (T6) and the third (T12) ocrelizumab infusions (Figure 1).

No pwMS received corticosteroids or other prescription or non-prescription drugs for at least 2 months prior to the start of ocrelizumab treatment. Finally, peripheral blood samples were also collected at a single time-point from age- and sex-matched healthy donors (HD) for baseline comparison.

Upon withdrawal, all samples were collected and centrifuged at 3000× *g* for 10 min to separate the plasma component, according to what the World Health Organization recommends [34]. Plasma samples were then stored at −80 °C until use.

### 2.3. Measurement of Plasma B-Cell-Activating Factors and Immunoglobulin Levels

Plasma levels of BAFF, APRIL and CD40L were assessed via the commercial cytometric bead-based multiplex panel immunoassay (CBA) LEGENDplex™ (BioLegend, San Diego, CA, USA), acquired using MACSQuant (Miltenyi Biotec, Bergisch Gladbach, Germany) and analyzed using FlowJo™ v10.8.1 software. B cell activation markers were expressed as plasma concentration (pg/mL).

As part of routine evaluation, baseline immunoglobulin (Ig) M, IgG and IgA plasma levels were assessed in all pwMS.

### 2.4. Statistical Analysis

Median values with interquartile ranges (IQR; 25th–75th percentiles) were reported for all quantitative data. The statistical analyses were carried out using GraphPad Prism 9. The median comparisons between pwMS and HD and between pwMS subgroups were performed via the non-parametric comparative Mann–Whitney test. Patient characteristics were compared using chi-square for categorical variables. The median longitudinal evaluations were performed via the non-parametric Friedman test. The median comparisons between pwMS and HD were performed via Dunn’s multiple comparison post-test. The correlations were obtained via the Spearman test.

In the analyses, *p* values below or equal to 0.05 (≤0.05) were considered statistically significant.

## 3. Results

### 3.1. Demographics and Clinical Characteristics of Study Population

From February 2018 to April 2021, 38 pwMS with RRMS (female/male: 14/24) with median age of 54 (47–61) years and 26 HD (female/male: 13/13) with median age of 52 (46–61) years were enrolled.

Among pwMS, the median time of disease was 11 (6–19) years and the median EDSS value was 5.5 (4.0–7.0). At enrollment, 34.2% (13/38) of pwMS were naïve to MS treatment. Among the non-naïve pwMS, the most common prior DMTs were fingolimod (24.0%), dimethyl fumarate (16.0%), IFN-β (12.0%) and natalizumab (12.0%).

The demographic and clinical characteristics of the ocrelizumab subpopulation are reported in Table 1.

### 3.2. Evaluation of Plasma BAFF, APRIL and CD40L Levels

At baseline, significantly higher plasma BAFF, APRIL and CD40L levels were observed in pwMS than in HD (*p* < 0.0001, *p* = 0.0223 and *p* < 0.0001, respectively) (Figure 2, Table 2).

At baseline, no differences in plasma BAFF, APRIL and CD40L levels were observed between non-naïve and naïve pwMS (BAFF: 739 [491–1257] and 1111 [494–1596], respectively; APRIL: 748 [465–1686] and 649 [321–1357], respectively; CD40L: 1111 [6432–2554] and 1063 [466–1601], respectively). Finally, no association between pre-treatment and infectious events was observed. Indeed, the occurrence of infectious events was 32% (8/25) in non-naïve pwMS and 46.2% (6/13) in naïve pwMS.

In pwMS, plasma BAFF, APRIL and CD40L levels were assessed longitudinally throughout T0, T6 and T12. The longitudinal values were then compared to HD. An overview of the plasma level evaluations of BAFF, APRIL and CD40L in ocrelizumab-treated pwMS and HD is reported in Table 2.

The longitudinal evaluation showed a significant increase in plasma BAFF levels at T12 compared to T0 (*p* < 0.0001), with a steep incremental rise between T0 and T6 (*p* < 0.0001) and a less marked but significant increase between T6 and T12 (*p* = 0.0026) (Figure 3A, Table 2). Upon comparison with HD, significant elevations in the plasma BAFF levels of pwMS were consistently observed at both T6 and T12 (*p* < 0.0001and *p* < 0.0001, respectively) (Figure 3A, Table 2).

On the other hand, in pwMS, plasma APRIL levels significantly decreased at T12 compared to T0 (*p* = 0.0003), as well as at T12 compared to T6 (*p* = 0.0250) (Figure 3B, Table 2). No significant differences in plasma APRIL levels at both T6 and T12 compared to HD were observed (Figure 3B, Table 2).

Likewise, significant decreases in plasma CD40L levels were observed in pwMS at T0 compared to T12 (*p* < 0.0001), as well as at T12 compared to T6 (*p* = 0.0124) (Figure 3C, Table 2). Higher plasma CD40L levels were seen in pwMS at T6 compared to HD (*p* = 0.0035) (Figure 3C, Table 2). No significant differences in plasma CD40L levels at T12 compared to HD were observed (Figure 3C, Table 2).

### 3.3. Correlation between B Lymphocyte Activation Markers and Immunoglobulin Levels

At T0, correlations between Ig plasma levels and B cell activation markers were investigated in pwMS. Positive correlations were identified between plasma levels of APRIL and IgG (Spearman ρ = 0.3847 and *p* = 0.0432) (Figure 4A), as well as between plasma levels of CD40L and IgG (Spearman ρ = 0.5348 and *p* = 0.0034) (Figure 4B).

### 3.4. Infectious Events and Plasma BAFF, APRIL and CD40L Levels

During the 12-month follow-up period, 14 pwMS out of 38 developed clinically significant infectious events. The infectious conditions reported included HSV-1 reactivations, recurrent urinary tract infections (UTIs) and respiratory tract infections (RTIs) (Table 3). In all pwMS that developed infectious events, specific antiviral or antibiotic treatment was required. Hence, the 38 pwMS were stratified in two subgroups: with and without infectious events (Table 3).

In the pwMS subgroups (with and without infectious event), plasma levels of BAFF, APRIL and CD40L were evaluated across the three time-points. In the with infectious event subgroup, significantly higher plasma BAFF levels were observed at all time-points compared to the without subgroup (T0: 1391 [1241–1841] and 576 [427–801] pg/mL, respectively, *p* < 0.0001; T6: 1782 [1506–2569] and 1162 [807–1794] pg/mL, respectively, *p* = 0.0056; T12: 2301 [1896–2546] and 1647 [1143–2420] pg/mL, respectively, *p* = 0.0400) (Figure 5A). Conversely, no significant differences were observed in plasma APRIL and CD40L levels between the two subgroups (Figure 5B,C). Similarly, at baseline, no differences in plasma Ig levels were observed (IgG: 11.3 [7.4–14.6] and 8.9 [8.1–12.3] g/L, respectively; IgA: 2.4 [1.6–3.7] and 2.2 [1.1–2.5] g/L, respectively; IgM: 1.0 [0.7–1.3] and 1.1 [0.8–1.3] g/L, respectively).

## 4. Discussion

The development of effective DMTs has considerably changed the management of MS with a deep positive impact on patients’ prognoses, annual relapse rates, disability progression and, most importantly, quality of life [7,35,36]. These advancements unraveled over the past three decades and much is yet to be investigated about their long-term effects and possible adverse events through real-life active monitoring and observation [36,37]. Although highly effective, DMTs carry an array of adverse effects [38,39] inducing patients to delay or discontinue MS treatment. Accordingly, the induction of varying degrees of immunosuppression in DMT-treated pwMS has been associated with an increased infectious risk, secondary to newly acquired or latent pathogens [31,40,41,42].

Despite the fact that chronic inflammation associated with MS was primarily considered a T-cell-mediated process, there is increasing evidence of the important role of B cells and humoral immunity in the genesis of demyelinating lesions [43]. Indeed, B cells are implicated in MS pathogenesis [44,45] by producing antibodies against myelin sheaths and axons [10,11]. However, an increased number of studies focused on further potential B cell functions, independently from the antibody–complement pathway, show that these have a relevant role in the production of inflammatory mediators [43] and cytotoxic molecules [46].

Immunotherapy targeting B cell populations has been found to slow disease progression, suggesting the role of B cells in both pathogenesis and progression [11,47]. However, compared with other B cell specific therapies, such as anti-CD20, atacicept targets BAFF and APRIL, cytokines mainly involved in B cell differentiation. Hence, anti-BAFF/APRIL agents do not further deplete B cell subsets as B cell progenitors and memory B cells that may play a substantial role in MS pathogenesis. Furthermore, the targeting of cytokines such as BAFF and APRIL might disrupt regulatory B cell pathways, which, in turn, could modulate T cell responses, thereby creating a proinflammatory environment and leading to an increase in relapses.

Finally, B cells are involved in the response to infections [48,49] and vaccination [50,51]. Thus, in the present study, the B cell activation markers BAFF, APRIL and CD40L were analyzed longitudinally in pwMS treated with ocrelizumab, correlating these findings to the occurrence of infectious events.

At baseline, higher levels of all three cytokines compared with HD were observed. These findings are consistent with previous reports on pwMS as well as reports on several autoimmune and inflammatory states [16,17,18,19,20,52,53,54]. Specifically, DMTs reportedly influence plasma BAFF levels in pwMS, but the significance of these changes remains unclear.

Upon longitudinal evaluation of the cytokine plasma levels in pwMS, plasma BAFF levels significantly increased across all time-points, whereas reduction in plasma APRIL and CD40L were observed. These temporal dynamic patterns may be reconducted to their different roles and activity on various B cell maturation phases [55,56,57]. BAFF specifically induces survival and maturation of naïve B cells, promoting B cell homeostasis [56,58]. During anti-CD20 treatment, the majority of cells that respond to BAFF stimulation are depleted [59]; thus, the increase in plasma BAFF levels could be due to either the lack of cells with receptors for BAFF, especially BAFF-R, or due to a feedback mechanism with the purpose of promoting B cell repopulation [57,60,61,62,63]. Moreover, MS induces a state of constant and intense immunoactivation, which may lead to immune dysfunction and, eventually, premature exhaustion [8,64]. In these conditions, BAFF is involved in the vicious cycle of persistent immunoactivation and may be uncontrollably increased [17]. The increase in plasma BAFF during anti-CD20 therapy has been documented in several reports about Systemic Lupus Erythematosus (SLE), Rheumatoid Arthritis (RA), Sjögren syndrome and, recently, in MS too [57,60,61,62,63].

On the other hand, as APRIL and CD40L are mostly involved in the antigen-dependent phases of B cell maturation and in the formation and maintenance of germinal centers [55,65,66], their plasma levels are reduced if such processes are blunted by anti-CD20 therapy [62]. Similar reductions in APRIL have also been observed in SLE patients after rituximab treatment [62]. Moreover, in this study, a positive correlation was observed between IgG levels and the plasma levels of APRIL and CD40L, respectively. This finding confirms the involvement of APRIL and CD40L in the germinal center reactions that lead to antibody synthesis, antigen affinity development and Ig isotype switch [66,67,68,69].

Furthermore, the association between B cell activation markers and the infectious risk in ocrelizumab-treated pwMS was evaluated. In the with infectious event subgroup, significantly higher plasma levels of BAFF were observed at baseline, and this incremental trend was retained throughout the first 12 months of treatment, compared to the without infectious event subgroup. This finding may be ascribed to the fact that pwMS with marked BAFF elevations may have more profound immune dysregulation [70,71,72]. Similar findings can be observed in subjects with persistent infections and inflammatory states [67,73]. In individuals suffering from chronic hepatitis B or C, higher plasma levels of BAFF during the acute phase of the disease have been correlated with disease severity, and the increase in B cell activation could promote the development of the dysregulated autoimmune phenomena associated with HCV [67,73]. Additionally, the significant increase in BAFF plasma levels observed in pwMS may be further augmented in the setting of infectious complications [67,74].

The limitations of our study include the lack of longitudinal evaluation of plasma Ig levels in the pwMS cohort. Moreover, prolonging the follow-up could be useful to better understand and potentially confirm our observations in pwMS during ocrelizumab treatment.

## 5. Conclusions

In conclusion, in ocrelizumab-treated pwMS, an increase in plasma BAFF levels and, over time, a reduction in plasma APRIL and CD40L levels are present. Furthermore, pwMS with high plasma levels of BAFF might have a more severe immune dysfunction and an ineffective immune response against community-acquired pathogens. Therefore, in this setting, BAFF might have a predictive role of immune dysfunction and, consequently, increase infection risk in pwMS.

## Figures and Tables

**Figure 1 biology-12-00587-f001:**
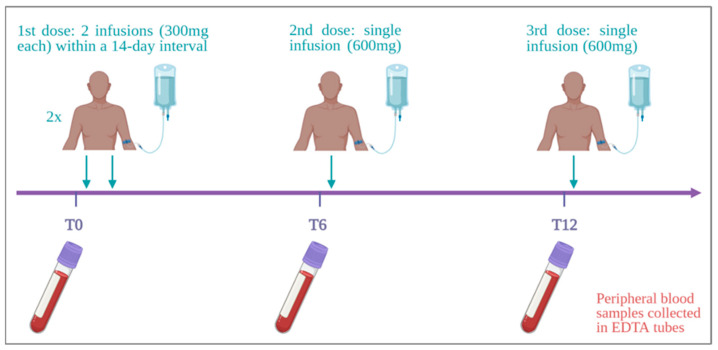
Timeline of ocrelizumab administration and concomitant peripheral blood sample withdrawal in ocrelizumab-treated pwMS. Ocrelizumab was administered within a 0.9% sodium chloride solution and the infusions were performed on a 6-month schedule. The first ocrelizumab dose was subdivided into two separate infusions (300 mg in 250 mL each), administered within a 14-day interval, whereas the subsequent doses were prepared as single infusions (600 mg in 500 mL). EDTA: ethylenediaminetetraacetic acid; T0: before first infusion; T6: before second infusion; T12: before third infusion.

**Figure 2 biology-12-00587-f002:**
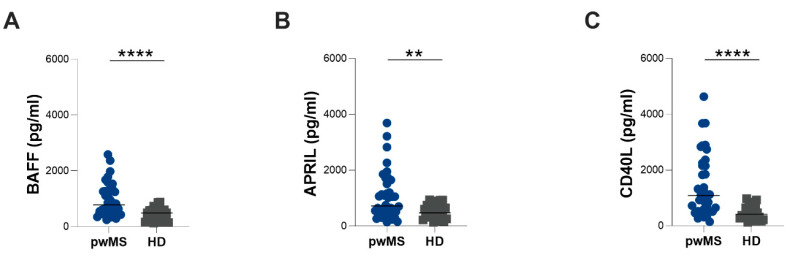
Baseline evaluation of plasma BAFF, APRIL and CD40L levels in pwMS. (**A**) Comparison of plasma BAFF levels between pwMS compared to HD. (**B**) Comparison of plasma APRIL levels in pwMS compared to HD. (**C**) Comparison of plasma CD40L levels at T0 in pwMS compared to HD. The non-parametric comparative Mann–Whitney test was used to compare medians between pwMS and HD. BAFF: B-cell-activating factor; APRIL: a proliferation-inducing ligand; CD40L: CD40 ligand; pwMS: people with MS; HD: healthy donors. **: *p* < 0.01; ****: *p* < 0.0001.

**Figure 3 biology-12-00587-f003:**
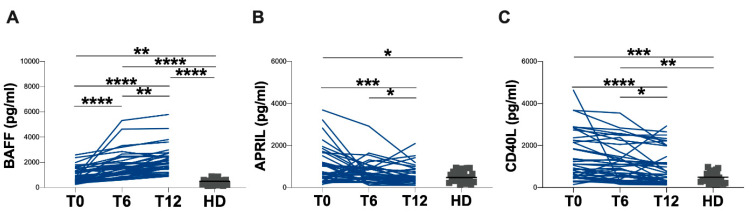
Longitudinal evaluation of plasma BAFF, APRIL and CD40L levels in pwMS and comparison with HD. (**A**) Longitudinal evaluation of plasma BAFF levels at T0, T6 and T12 in pwMS, as well as comparison with HD. (**B**) Longitudinal evaluation of plasma APRIL levels at the three time-points in pwMS, as well as comparison with HD. (**C**) Longitudinal evaluation of plasma CD40L levels at T0, T6 and T12 in pwMS, as well as comparison with HD. BAFF: B-cell-activating factor; APRIL: a proliferation-inducing ligand; CD40L: CD40 ligand; pwMS: people with MS; HD: healthy donors; T0: before first infusion; T6: before second infusion; T12: before third infusion. *****
*p* < 0.05; ******
*p* < 0.01; *******
*p* < 0.001; ********
*p* < 0.0001.

**Figure 4 biology-12-00587-f004:**
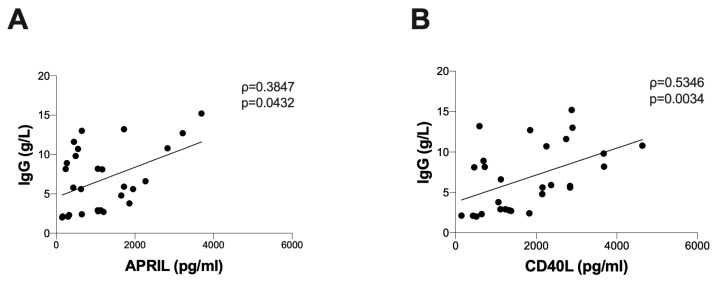
Correlations between plasma IgG and APRIL and CD40L levels. (**A**) Positive correlation between APRIL and IgG plasma levels. Linear correlation was evaluated by using the regression test, R^2^ = 0.2285 *p* = 0.0101. (**B**) Positive correlation between plasma CD40L and IgG levels. Linear correlation was evaluated by using the regression test, R^2^ = 0.1975 *p* = 0.0178. All correlations were performed using the Spearman test. The Spearman coefficient (ρ) and statistical significance (p) are reported in the graphics. IgG: immunoglobulin G; APRIL: a proliferation-inducing ligand; CD40L: CD40 ligand.

**Figure 5 biology-12-00587-f005:**
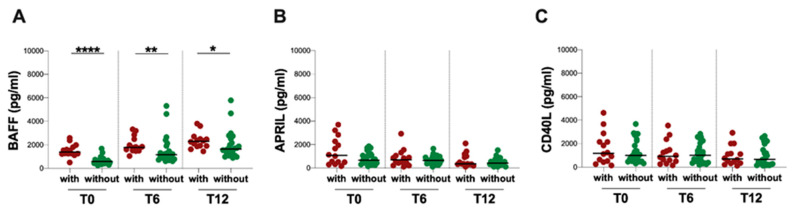
Evaluation of plasma BAFF, APRIL and CD40L levels in pwMS stratified according to the occurrence of infectious event. (**A**) Evaluation of plasma BAFF levels at T0, T6 and T12 between the with and without infectious event subgroups. (**B**) Evaluation of plasma APRIL levels at T0, T6 and T12 between the with and without infectious event subgroups. (**C**) Evaluation of plasma CD40L levels at T0, T6 and T12 between the with and without infectious event subgroups. BAFF: B-cell-activating factor; APRIL: a proliferation-inducing ligand; CD40L: CD40 ligand; pwMS: people with MS; T0: before first infusion; T6: before second infusion; T12: before third infusion. * *p* < 0.05; ** *p* < 0.01; **** *p* < 0.0001.

**Table 1 biology-12-00587-t001:** Clinical and demographic features of enrolled pwMS.

	pwMS (*n* = 38)
Female/Male	14/24
Age, median (IQR)	54 (47–61)
Years of disease, median (IQR)	11 (6–19)
EDSS, median (IQR)	5.5 (4.0–7.0)
Prior treatment	
alemtuzumab	1
azathioprine	2
daclizumab	1
dimethyl fumarate	4
fingolimod	6
glatiramer acetate	2
IFN-β	3
natalizumab	3
rituximab	1
teriflunomide	2
none	13
Plasma Ig levels	
IgG (g/L)	5.8 (2.8–10.5)
IgA (g/L)	2.0 (1.6–2.7)
IgM (g/L)	0.9 (0.8–1.2)

IQR: interquartile range; EDSS: Expanded Disability Status Scale; pwMS: people with MS; IFN-β: interferon-βeta; Ig: immunoglobulin.

**Table 2 biology-12-00587-t002:** Longitudinal evaluation of plasma BAFF, APRIL and CD40L levels in pwMS and comparison HD.

	pwMS				HD			
	T0	T6	T12	*p* ^§^		*p* ^†^	*p* ^††^	*p* ^†††^
BAFF (pg/mL)	781 (495–1288)	1462 (1060–2011)	1891 (1506–2497)	<0.0001	494 (211–611)	0.0073	<0.0001	<0.0001
APRIL (pg/mL)	724 (443–1548)	642 (407–919)	383 (319–791)	0.0004	477 (303–751)	0.0223	ns	ns
CD40L (pg/mL)	1087 (519–2281)	965 (474–1781)	665.50 (298–1599)	<0.0001	417 (298–724)	0.0002	0.0035	ns

Plasma BAFF, APRIL and CD40L levels were longitudinally evaluated in pwMS at T0, T6 and T12. **^§^**: The non-parametric comparative Friedman test was used to compare medians between T0, T6 and T12. The longitudinal plasma BAFF, APRIL and CD40L values of pwMS were compared to the levels displayed by HD. **^†^**: Dunn’s multiple comparison post-test was used for comparing medians between T0 and HD; **^††^:** Dunn’s multiple comparison post-test was used for comparing medians between T6 and HD; **^†††^**: Dunn’s multiple comparison post-test was used for comparing medians between T12 and HD. pwMS: people with MS; HD: healthy donors; BAFF: B-cell-activating factor; APRIL: a proliferation-inducing ligand; CD40L: CD40 ligand; T0: before first infusion; T6: before second infusion; T12: before third infusion.

**Table 3 biology-12-00587-t003:** Clinical and demographic features of pwMS stratified according to the occurrence of infectious event.

	pwMS (*n* = 38)	
	With Infectious Event (*n* = 14)	Without Infectious Event (*n* = 24)
Female/Male	3/11	11/13
Age, median years [IQR]	60 (50–65)	51 (47–56)
Years of disease, median (IQR)	17 (11–23)	9 (5–14)
EDSS, median (IQR)	6.5 (5.5–7)	5 (3–6)
Infectious event:		
HSV-1 reactivation	2	-
UTIs	9	-
RTIs	5	-
Antiviral/antibiotic treatment	14	-

IQR: interquartile range; pwMS: people with MS; EDSS: Expanded Disability Status Scale; HSV-1: Herpes simplex virus 1; UTIs: urinary tract infections; RTIs: respiratory tract infections.

## Data Availability

The original contributions presented in the study are included in the article. Further inquiries can be directed to the corresponding author.

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
