# Peer review of "Evaluation of BAFF, APRIL and CD40L in Ocrelizumab-Treated pwMS and Infectious Risk"

_biology, 2023, doi:10.3390/biology12040587_

Round 1

Reviewer 1 Report

The manuscript "Evaluation of BAFF, APRIL and CD40L in ocrelizumab-treated pwMS and infectious risk" by Maria Antonella Zingaropoli et al. investigates the relationship of B-cell activating factors plasma levels and infectious risk in ocrelizumab-treated MS patients at three time points. Additional useful information was obtained by dividing and following patients with or without an infectious event.

The manuscript is well organized, and the content is in accordance with the title. The measurement of factors of interest in blood samples from MS patients and healthy donors were performed and statistically analysed by appropriate methods. The results are well structured and clearly presented. The references are appropriate and up to date. The conclusions contain information useful to the clinicians regarding the BAFF plasma levels and immune dysfunction, and potentially increased infectious risk in MS patients. Considering all the above, I believe that the manuscript will be interesting and useful to the scientific public of the Journal.

Reviewer 2 Report

Very important article with clinical consequences. AntiCD20 therapy is more and more popular but it has the potential to induce infections in some patients and is able to lower IgG levels.

Though the sample size is small and especially the part of patients with infections is extremely small, the results are very encouraging and need confirmation in much larger sample.

The only problem is that infections do not usually develop in the first year of the treatment. The IgG levels decrease also does not develop abruptly. Therefore it would be better to prolong the follow up and confirm that the mechanism which is behind infections and lower IgG levels is working also later. It would be more meaningful.

But in fact, I see this as the only limitation which should be mentioned in the conclusions.

Author Response

Response to Reviewer 2 Comments

Very important article with clinical consequences. AntiCD20 therapy is more and more popular but it has the potential to induce infections in some patients and is able to lower IgG levels.

Though the sample size is small and especially the part of patients with infections is extremely small, the results are very encouraging and need confirmation in much larger sample.

The only problem is that infections do not usually develop in the first year of the treatment. The IgG levels decrease also does not develop abruptly. Therefore it would be better to prolong the follow up and confirm that the mechanism which is behind infections and lower IgG levels is working also later. It would be more meaningful.

But in fact, I see this as the only limitation which should be mentioned in the conclusions.

As suggested by the Referee #2, in the “Discussion” section” (page 9, lines 329-331), we mentioned that prolonging the follow up could be useful to better understand and confirm our observations in pwMS during ocrelizumab treatment.

Reviewer 3 Report

In this article, Zingaropoli et al. investigated a cohort of 38 MS-patients on CD 20 depletion longitudinally. They found both significant differences in BAFF, APRIL and CD40L compared to healthy controls, and an increase in BAFF, and a decrease in APRIL and CD40L within 12 months. Furthermore, dividing the group of patients according to a subgroup with and without infections, the authors found higher BAFF levels in the group with infectious events.

This article highlights the role of B-cell activation markers in the disease monitoring during CD20 therapy. The topic is highly important, the manuscript is well written, resulting in an article that is worth publication in this journal. I only have a few comments:

1. The authors presented the results of the Ig levels at baseline. Did the authors also collect the Ig levels longitudinally? Did they differ between the   infectious/non-infectious group? Please discuss.

2. What about the status of prior treatment? Did the group with and without prior treatment differ in BAFF/APRIL/CD40L? Was there an association between pre-treatment and infections? Please discuss. 

3. When did the infections occur? Is there any information about that? Can the authors conclude that an increased level of BAFF is associated with an infectious event, or that an infectious event is associated with an increased level of BAFF? Please discuss.

4. Any explanation why the two patients with the highest level of BAFF were in the non-infectious group? Please discuss.

5. Can the authors discuss the literature about the failure of atacicept in the treatment of MS?

In sum, I recommend minor revision. 

Author Response

Response to Reviewer 3 Comments

In this article, Zingaropoli et al. investigated a cohort of 38 MS-patients on CD 20 depletion longitudinally. They found both significant differences in BAFF, APRIL and CD40L compared to healthy controls, and an increase in BAFF, and a decrease in APRIL and CD40L within 12 months. Furthermore, dividing the group of patients according to a subgroup with and without infections, the authors found higher BAFF levels in the group with infectious events.

This article highlights the role of B-cell activation markers in the disease monitoring during CD20 therapy. The topic is highly important, the manuscript is well written, resulting in an article that is worth publication in this journal. I only have a few comments:

  1. The authors presented the results of the Ig levels at baseline. Did the authors also collect the Ig levels longitudinally? Did they differ between the infectious/non-infectious group? Please discuss.

Unfortunately, we did not collect plasma Ig levels longitudinally (we reported this aspect as limitation of the study, page 9, line 328). However, as suggested by the Referee #3, we performed a comparison of plasma Ig levels at baseline by stratifying pwMS according to the occurrence of the infectious event. As reported in the "Results" section (page 7, lines 259-262), no differences were found in plasma Ig levels at baseline.

  1. What about the status of prior treatment? Did the group with and without prior treatment differ in BAFF/APRIL/CD40L? Was there an association between pre-treatment and infections? Please discuss.

As suggested by the Referee #3, we evaluated the differences in plasma levels of BAFF, APRIL and CD40L between the naïve and non-naïve pwMS. However, as reported in the “Results” section (page 5, lines 186-189), at baseline no differences were found. Finally, using chi-square for categorical variables (we implemented “Materials and Methods” section, page 3, lines 126-127), no association between pre-treatment and infectious event was observed (32% of non-naïve pwMS with infectious event versus 46.2% naïve pwMS with infectious event, p=0.4664) (page 5, lines 189-191).

  1. When did the infections occur? Is there any information about that? Can the authors conclude that an increased level of BAFF is associated with an infectious event, or that an infectious event is associated with an increased level of BAFF? Please discuss.

As previously described (Zingaropoli et al. 2022), in the context of a collaboration between the MS Centre and the Neuroinfectious Unit, all pwMS are evaluated every six months to recognize new or reactivation of latent infections. However, pwMS can contacted us because of infection event and they are evaluated at the Neuroinfectious Unit ahead of time. From our observations, we can hypothesize that an infectious event is associated with increased plasma BAFF levels. In fact, several studies (Rodriguez et al. 2003; Toubi et al. 2006; Yang et al. 2014) have shown that BAFF expression is stimulated by most microbes through their pathogen-associated molecular patterns, directly involving molecules such as TLRs or inducing type I and type II interferons.

  1. Any explanation why the two patients with the highest level of BAFF were in the non-infectious group? Please discuss.

The higher plasma levels of BAFF could reflect an increased state of immune activation, as this factor acts on several immune cells, including T cells, B cells, and dendritic cells. The reason for the higher plasma levels of BAFF in two pwMS in the without infectious group is not well understood by us. Further in-depth studies might be helpful to better understand the involvement of BAFF in MS.

  1. Can the authors discuss the literature about the failure of atacicept in the treatment of MS?

As suggested by the Referee #3, in “Introduction” (page 2, lines 64-75) and “Discussion” sections (page 8, lines 276-284), we discussed the literature about the failure of atacicept in MS treatment.

In sum, I recommend minor revision.

Reviewer 4 Report

The authors studied the effect of three doses of ocrelizumab on plasma markers, Ig, and infectious events in people with MS. The third dose came after the final blood sample so could not have affected the blood measures. Of the people with MS, 13 had no prior treatment and the rest had various standard drug treatments. A healthy donor age and sex-mathced control group was used for baseline comparisons. The mansucript was very clear and the figures were easy to follow, with appropriate statistics and interpretations. Considering how widely used this drug has become, the mansucript should be of interest to neurologists, immunologists, and people with MS.

Comments

Corticosteroids was not mentioned.  The patients should not have received corticosteroids for at least 2 months prior to the study to avoid confounding effects. Were there any other exclusion factors such as co-morbitities, other prescription/non-prescription drugs, BMI,  etc, if so explain in methods.  

The intro and methods should make clear what type of patients are being studied (presumably RRMS), and line 74 should specify how many were recruited.

Table 2 caption, the use of * is unclear, should use symbol other than * , it looks like a p value, but its more of a label.

Its not clear how infectious events were recorded. Did people with MS go to doctor when they were sick, or did they go for regular scheduled visits  in the 12 month follow-up? It is important to explain the procedure of the 12 months follow-up in more detail so that the possible sampling bias can be assessed. If there were regular visits, how was the compliance in the patient group during the 12 month follow-up?

There was no BAFF/CD40L/APRIL comparison between the 13 people with MS who were treatment-naive as compared to people with MS who were taking drug treatments as a subgroup. Even if there is no significant difference, it would be important to report this comparison in supplemental.

T g force reported for plasma preparation (300 g) seems low compared to other protocols, on ThermoFisher for example they recommend 1000-2000g. But there seems to be variations in protocols. Authors could check that, and provide a reference for their protocol. Was the centrifuge refrigerated?

The paper was very well written, just some grammar issues on line 52/53 , 64, 237, 240, 282

Author Response

Response to Reviewer 4 Comments

The authors studied the effect of three doses of ocrelizumab on plasma markers, Ig, and infectious events in people with MS. The third dose came after the final blood sample so could not have affected the blood measures. Of the people with MS, 13 had no prior treatment and the rest had various standard drug treatments. A healthy donor age and sex-mathced control group was used for baseline comparisons. The mansucript was very clear and the figures were easy to follow, with appropriate statistics and interpretations. Considering how widely used this drug has become, the mansucript should be of interest to neurologists, immunologists, and people with MS.

Comments

Corticosteroids was not mentioned.  The patients should not have received corticosteroids for at least 2 months prior to the study to avoid confounding effects. Were there any other exclusion factors such as comorbitities, other prescription/non-prescription drugs, BMI, etc, if so explain in methods. 

As suggested by Referee #4, we specified in the "Materials and Methods" section that no patients had received corticosteroids or other prescription or nonprescription drugs for at least 2 months prior to the start of treatment with ocrelizumab (page 3, lines 105-106).

The intro and methods should make clear what type of patients are being studied (presumably RRMS), and line 74 should specify how many were recruited.

As suggested by Referee #4, we specified that all pwMS had relapsing-remitting form (page 4, line 135). We prefer to report the number of patients enrolled in the "Results" section since this is an outcome of the study (when we started the study, we did not know the number of patients we would include).

Table 2 caption, the use of * is unclear, should use symbol other than * , it looks like a p value, but its more of a label.

As suggested by Referee #4, we used other symbols (§ and ) in Table 2.

Its not clear how infectious events were recorded. Did people with MS go to doctor when they were sick, or did they go for regular scheduled visits  in the 12 month follow-up? It is important to explain the procedure of the 12 months follow-up in more detail so that the possible sampling bias can be assessed. If there were regular visits, how was the compliance in the patient group during the 12 month follow-up?

As previously described (Zingaropoli et al. 2022), in the context of a collaboration between the MS Centre and the Neuroinfectious Unit, all pwMS are evaluated every six months to recognize new or reactivation of latent infections. In case of increased infectious risk or active infectious diseases needing a specific treatment, an official report is sent to the MS Centre to inform the reference neurologist and discuss about the therapeutic approach. In the present study, all pwMS underwent regular visits every six months and to avoid sampling bias each sample was taken before ocrelizumab infusion.

There was no BAFF/CD40L/APRIL comparison between the 13 people with MS who were treatment-naive as compared to people with MS who were taking drug treatments as a subgroup. Even if there is no significant difference, it would be important to report this comparison in supplemental.

As suggested by Referee #4, we performed a comparison between naïve and non-naïve pwMS in plasma BAFF, APRIL and CD40L (page 5, lines 186-189).

T g force reported for plasma preparation (300 g) seems low compared to other protocols, on ThermoFisher for example they recommend 1000-2000g. But there seems to be variations in protocols. Authors could check that, and provide a reference for their protocol. Was the centrifuge refrigerated?

Thanks to the suggestion of Referee #4, we corrected the plasma separation method (it was a typo) as follows “Upon withdrawal, all samples were collected and centrifuged at 3000xg for 10 minutes to separate the plasma component, according to the World Health Organization recommends (World Health Organization. Diagnostic Imaging and Laboratory Technology 2002). Plasma samples were then stored at -80°C until use” (page 3, lines 108-111). According to the manufacturer's instructions, the centrifuge used was not refrigerated.

The paper was very well written, just some grammar issues on line 52/53 , 64, 237, 240, 282

Thanks to the suggestion of Referee #4, we revised the grammar issues as following:

Line 52-53: “The growing understanding of the immunopathogenesis of multiple sclerosis (MS) has led to the development of innovative treatments, and disease-modifying therapies (DMTs) have become the gold standard” (page 2, lines 51-53)

Line 64: “Finally, despite its remarkable efficacy, ocrelizumab is associated with an increased risk of infection and reactivation” (page 2, lines 76-77)

Line 237: “B cells are implicated in MS pathogenesis (Pröbstel, Sanderson, e Derfuss 2015; Krumbholz e Meinl 2014) by producing antibodies against myelin sheaths and axons (Kappos et al. 2011; Hauser et al. 2017). Immunotherapy targeting B cell populations has been found to slow disease progression. However, compared with other B cell specific therapies, such as anti-CD20, atacicept targets BAFF and APRIL, cytokines mainly involved in B cell differentiation. Hence, anti-BAFF/APRIL agents do not deplete further B cell subsets, as B cell progenitors and memory B cells that may play a substantial role in MS pathogenesis. Furthermore, targeting of cytokines such as BAFF and APRIL might disrupt regulatory B cell pathways, which in turn could modulate T cell responses, thereby creating a proinflammatory environment and leading to an increase in relapses. Finally, B cells are involved in responses to infections (Lam, Smith, e Baumgarth 2020; Elsner e Shlomchik 2020) and vaccination (Iannetta et al. 2021; Dominelli et al. 2022). Thus, in the present study, the B cell activation markers, BAFF, APRIL, and CD40L, were analyzed longitudinally in pwMS treated with ocrelizumab, correlating these findings to the occurrence of infectious events.” (page 8, lines 276-288)

Line 240: “At baseline, higher levels of all three cytokines compared with HD were observed” (page 8, line 289)

Line 282: “Furthermore, pwMS with high plasma levels of BAFF might have severe immune dysfunction and an ineffective immune response against community-acquired pathogens. Therefore, in this setting, BAFF might have a predictive role of immune dysfunction and consequent increased infection risk in pwMS” (page 9, lines 334-338).